# Urban Feature Extraction within a Complex Urban Area with an Improved 3D-CNN Using Airborne Hyperspectral Data

Xiaotong Ma [1], Qixia Man [1,*], Xinming Yang [2], Pinliang Dong [3], Zelong Yang [1], Jingru Wu [1] and Chunhui Liu [1]

[1] College of Geography and Environment, Shandong Normal University, Jinan 250014, China
[2] Jinan Environmental Research Institute, Jinan 250000, China
[3] Department of Geography and the Environment, University of North Texas, Denton, TX 76203, USA
*   Correspondence: qixiaman@sdnu.edu.cn; Tel.: +86-15665876268

**Abstract:** Airborne hyperspectral data has high spectral-spatial information. However, how to mine and use this information effectively is still a great challenge. Recently, a three-dimensional convolutional neural network (3D-CNN) provides a new effective way of hyperspectral classification. However, its capability of data mining in complex urban areas, especially in cloud shadow areas has not been validated. Therefore, a 3D-1D-CNN model was proposed for feature extraction in complex urban with hyperspectral images affected by cloud shadows. Firstly, spectral composition parameters, vegetation index, and texture characteristics were extracted from hyperspectral data. Secondly, the parameters were fused and segmented into many S × S × B patches which would be input into a 3D-CNN classifier for feature extraction in complex urban areas. Thirdly, Support Vector Machine (SVM), Random Forest (RF),1D-CNN, 3D-CNN, and 3D-2D-CNN classifiers were also carried out for comparison. Finally, a confusion matrix and Kappa coefficient were calculated for accuracy assessment. The overall accuracy of the proposed 3D-1D-CNN is 96.32%, which is 23.96%, 11.02%, 5.22%, and 0.42%, much higher than that of SVM, RF, 1D-CNN, or 3D-CNN, respectively. The results indicated that 3D-1D-CNN could mine spatial-spectral information from hyperspectral data effectively, especially that of grass and highway in cloud shadow areas with missing spectral information. In the future, 3D-1D-CNN could also be used for the extraction of urban green spaces.

**Keywords:** airborne hyperspectral image; deep learning; 3D-CNN; pixel-based; 3D-1D-CNN

## 1. Introduction

Urban feature extraction based on hyperspectral data plays an important role in many applications, such as urban planning, change detection, and urban environmental monitoring [1–4]. Hyperspectral images (HSI) can provide tens or hundreds of high spectral resolution images covering the visible to the infrared region [5], which gives the possibility of fine classification of classes with spectral similarity [6]. However, the numerous bands [7] and the presence of cloud shadows [8] could reduce the classification accuracy significantly. Therefore, it is necessary and meaningful to fully exploit and utilize spatial-spectral information from a hyperspectral image in complex urban areas, especially in cloud shadow areas.

Urban feature extraction has always been a difficult and hot issue due to the problems of different things with the same spectrum patterns, the same things with different spectrums, shadows [9], and spatial heterogeneity [10]. Through high-resolution hyperspectral data to provide detailed structural information and spectral information, Chen et al. [11] drew an urban land cover map, and the overall classification accuracy reached 97.24%. Clarks et al. [12] compared the effect of hyperspectral characteristics in different seasons on the mapping of land cover. Compared with the summer hyperspectral index, the overall accuracy of the multitemporal hyperspectral index is improved by 0.9~3.1%. The appearance of hyperspectral images at different times with high resolution makes it

possible to overcome those problems with more details and improve the accuracy. Chen et al. [13] used hyperspectral data for urban feature extraction using SVM [14], and the overall accuracy is 90.02%. Zhang et al. [15] utilized hyperspectral data and LiDAR data for the extraction of urban tree species using RF and achieved an overall accuracy of 87.00%. Tamilarasi and Prabu [16] have extracted roads and buildings in urban areas using SVM, and the achieved accuracy is 78.34% and 92.47%, respectively. As for the shadowing problem, Qiao et al. [17] used hyperspectral data to classify all shaded pixels in urban areas with different land cover types using a maximum likelihood classifier (MLC) and SVM classifier. Luo et al. [18] separately classified the shaded and the unshaded areas and acquired the formal results by decision fusion with an overall accuracy of 95.92%. Man et al. [19] employed pixel-based support vector machines and object-based classifiers to extract urban objects in cloud shadows. Rasti et al. [20] applied orthogonal total variation component analysis (OTVCA) to urban hyperspectral images with high spatial resolution. Compared with random forests and support vector machines, the features extracted by OTVCA show considerable improvement in classification accuracy. To better explore the spectral characteristics of HSI, Zhang et al. [21] improved the simple linear iterative cluster (SLIC) method, which showed better classification performance on three HIS compared with SVM. However, these studies shown that traditional classification methods could not make full use of the rich spatial-spectral information of hyperspectral data. However, these studies shown that traditional classification methods could not make full use of the rich spatial-spectral information of hyperspectral data. Additionally, most of the studies classified the urban objects in shadow areas separately. Therefore, there is an urgent need for a method that could fully mine the spectral-spatial information of hyperspectral data, especially including the shadow areas.

With the development of remote sensing technology, the spatial and spectral resolution of hyperspectral data is getting higher and higher. How to make full use of spatial-spectral information is particularly important. Recently, the 3D-CNN algorithm could take the spectrum as the third dimension to fully utilize the spatial-spectral information of hyperspectral data. Chen et al. [22] extracted spectral features from one-dimensional CNN extracted local spatial features of each pixel from two-dimensional CNN, and further developed 3-D CNN to learn the spatial and spectral features of his. Many types of research have proved that 3D-CNN has good performance in handling hyperspectral data. Ying et al. [23] verified the performance of 3D-CNN on three hyperspectral datasets and achieved an accuracy of 95.00%. Nezami et al. [24] used 3D-CNN to classify tree species in the boreal forest and achieved 98.30%. However, as the dimensionality of the convolution kernel increases, the time cost also increases. Therefore, many researchers have improved 3D-CNNs and tested their performance in hyperspectral classification. Liu et al. [25] have proved that the 2D-3D-CNN hyperspectral classification method is better than that of 3D-CNN on three popular datasets. The main limitation of this method is the model is very complicated and has a lot of training parameters. The 3D-1D-CNN proposed by Zhang et al. [26] achieved 93.86% accuracy in classifying tree species in the artificial forest using all hyperspectral bands. This method could effectively reduce the complexity of the model while obtaining high classification accuracy. The 3D-1D-CNN not only overcomes the problem of spectral similarity between tree species, but also reduces the time through the structure improvement of the model. This timely, accurate, and lightweight deep learning model provides support for classification of complex and spectrally similar urban objects. However, these studies have been only experimented in artificial forests or agricultural fields, and has not been widely used in other land cover types.

Different from natural forests, urban objects have spectral and structural diversity. How about the performance of 3D-1D-CNN in complex urban areas? It is necessary to conduct tests in complex urban areas to validate the performance in previously poorly performed situations such as shadowed areas, hard-to-distinguish objects, and the varying spectrum patterns at different scenarios of 3D-1D-CNN. The main objective of this paper

is to evaluate the ability of the 3D-1D-CNN model in mining spectral-spatial information from hyperspectral data in complex urban areas, especially in cloud-shaded areas.

## 2. Datasets

### 2.1. Houston Dataset

The dataset includes an airborne hyperspectral image and ground truth data, which are available from the 2013 IEEE GRSS Data Fusion Contest (https://hyperspectral.ee.uh.edu/?page_id=459, (accessed on 16 March 2022). The reference data and the false color image of the dataset are presented in Figure 1.

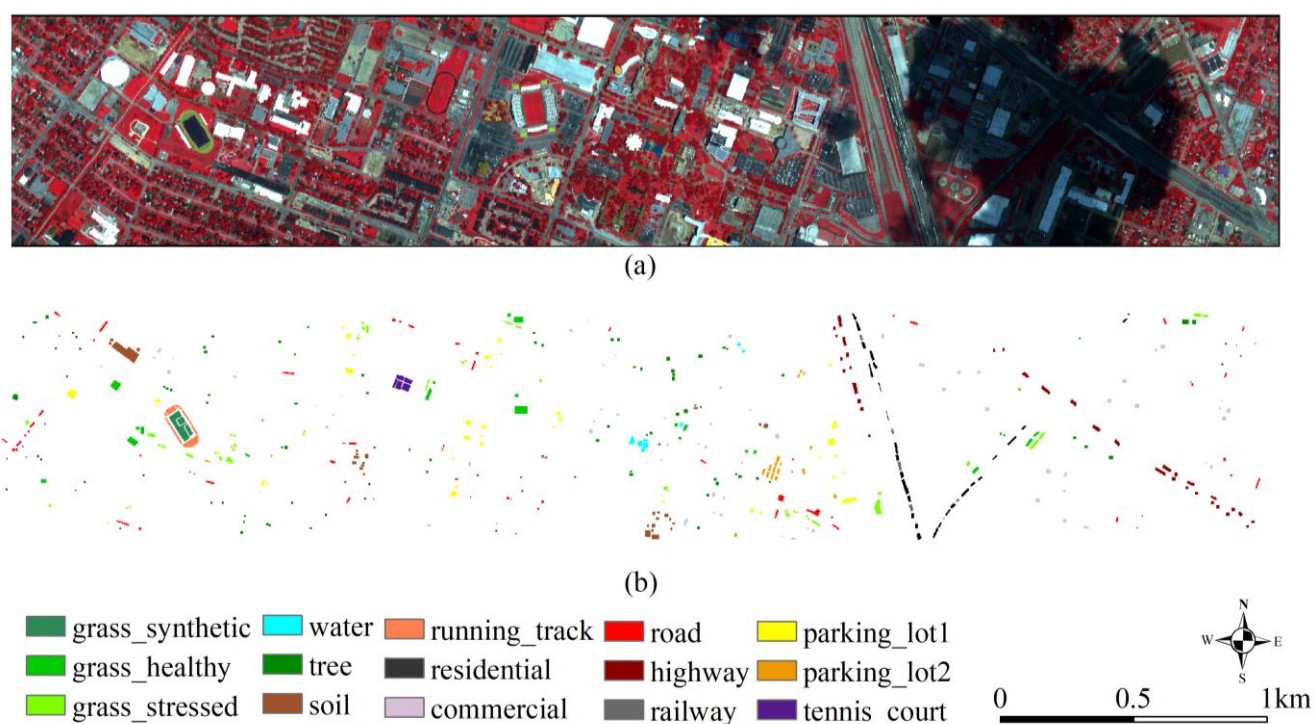

**Figure 1.** (**a**) The false-color image of the Houston dataset. (**b**) The ground truth of the Houston dataset.

The hyperspectral image was acquired by a CASI sensor on 23 June 2013, between the times 17:37:10 UTC and 17:39:50 UTC (Figure 1a). The data consists of 144 bands, and its spatial resolution and spectral resolution are 2.5 m and 4.8 nm, respectively. The spatial dimensions of the hyperspectral image are 1895 by 349 pixels.

There are great challenges in urban feature extraction using hyperspectral data in this area due to the following reasons: (1) Part of the dataset is seriously affected by cloud shadow, which will cause spectral loss of urban objects; (2) There is a problem with urban feature extraction caused by objects with different spectrums and objects with the similar spectrum; (3) Spectral-spatial information of hyperspectral data has not been fully mined; (4) Insufficient training sample or too much noise interference in the training data might cause the overfitting of the classification methods.

In this paper, 15 classes will be classified, including grass_healthy, grass_stressed, grass_synthetic, tree, soil, water, residential, commercial, road, highway, railway, parking_lot1, parking_lot2, tennis_court, and running_track. The number of training and validation samples is shown in Table 1, and Figure 1b demonstrates their distribution.

**Table 1.** Summary of training and validation samples of Houston dataset.

| Class | Sample | |
|---|---|---|
| | Train | Validation |
| commercial | 83 | 77 |
| grass_healthy | 84 | 28 |
| grass_stressed | 90 | 72 |
| grass_synthetic | 40 | 2 |
| highway | 88 | 50 |
| parking_lot1 | 88 | 54 |
| parking_lot2 | 28 | 28 |
| railway | 94 | 14 |
| residential | 91 | 122 |
| road | 98 | 84 |
| running_track | 41 | 10 |
| soil | 80 | 40 |
| tennis_court | 21 | 6 |
| tree | 77 | 106 |
| water | 10 | 14 |
| Total | 1013 | 707 |

### 2.2. Surrey Dataset

A hyperspectral image was acquired by a CASI sensor on 30 April 2013. The data consists of 72 bands with spatial and spectral resolutions of 1 m and 9.6 nm, respectively. The spatial dimension of the hyperspectral image is 1655 by 988 pixels. The false color image of the dataset is presented in Figure 2. The information of data sets is shown in Table 2

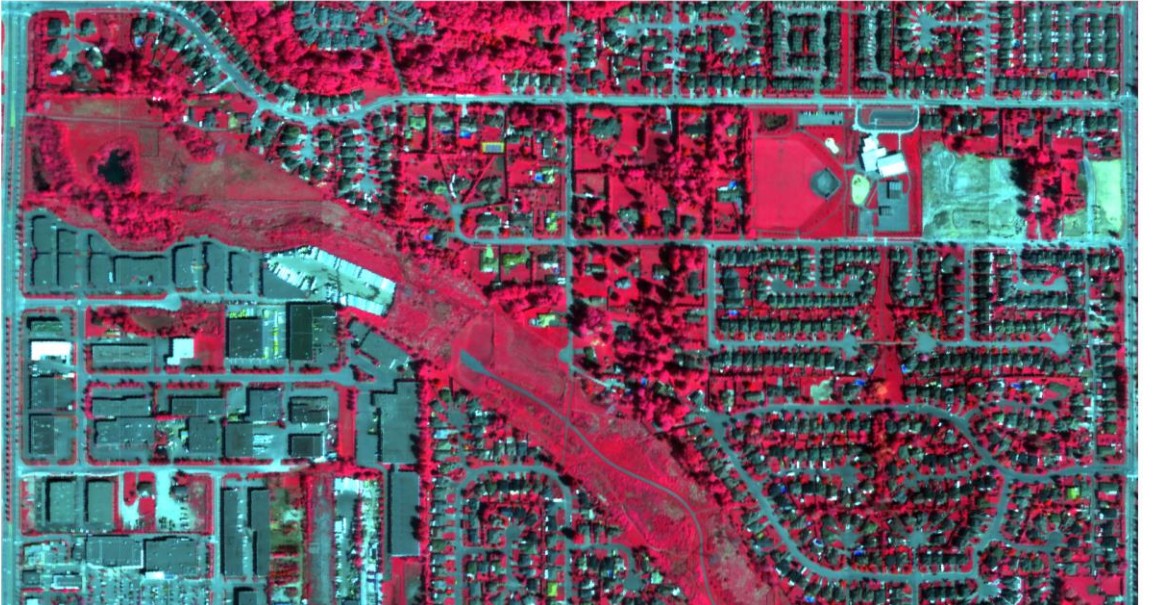

**Figure 2.** The hyperspectral image of the Surrey dataset in false-color display.

**Table 2.** Experimental dataset information.

| Dataset | Houston Dataset | Surrey Dataset |
|---|---|---|
| Sensor | CASI | CASI |
| Acquisition time | 2013/6/23 | 2013/4/30 |
| Number of Bands | 144 | 72 |
| Size | $1895 \times 349$ | $1655 \times 988$ |
| Number of Classes | 15 | 7 |
| Spatial resolution | 2.5 m | 1 m |
| Spectral resolution | 4.8 nm | 9.6 nm |

## 3. Methodology

### 3.1. Data Pre-Processing

For atmospheric correction, FLAASH is used to remove the influence of the atmosphere. Furthermore, the minimum noise fraction (MNF) [27] is employed to reduce or eliminate the Hughes phenomenon of hyperspectral data. The purpose of principal component analysis (PCA) [28] is to obtain the first principal component used to generate the texture characteristics. Additionally, the Gray-level Co-occurrence Matrix (GLCM) [29] and Normalized Difference Vegetation Index (NDVI) [30] are used to obtain texture characteristics as well as vegetation feature information. Table 3 shows the details of these variables.

**Table 3.** Variable and their respective descriptions and reference.

| Variable | Description | Reference |
|---|---|---|
| Spectral composition parameters | | |
| MNF22 | First 22 components of minimum noise fraction | [22] |
| Vegetation Index | | |
| Normalized Difference Vegetation Index (NDVI) | $NDVI = \frac{(IR-R)}{(IR+R)}$ | [25] |
| Texture characteristics | | |
| Mean | $\sum\limits_{i,j=0}^{N-1} i\left(P_{i,i}\right)$ | [24] |
| Variance | $\sum\limits_{i,i=0}^{N-1} P_{i,i}(i-\mu_i)^2 \sigma_j^2 = \sum\limits_{i,i=0}^{N-1} P_{i,i}(j-\mu_i)^2$ | [24] |
| Homogeneity | $\sum\limits_{i,j=0}^{N-1} \frac{P_{i,j}}{1+(i-j)^2}$ | [24] |
| Contrast | $\sum\limits_{k=0}^{N-1} \sum\limits_{i=-1}^{k} P_{i,j} \times k \times k$ | [24] |
| Dissimilarity | $\sum\limits_{k=0}^{N-1} \sum\limits_{|i-j|}^{k} P_{i,j} \times k$ | [24] |
| Entropy | $\sum\limits_{i,j=0}^{N-1} P_{i,j}\left(-\ln P_{i,i}\right)$ | [24] |
| Second Moment | $\sum\limits_{i,j=0}^{N-1} P_{i,j}{}^2$ | [24] |
| Correlation | $\sum\limits_{i,j=0}^{N-1} P_{i,j}\left[\frac{(i-\mu_i)(j-\mu_j)}{\sqrt{(\sigma_i^2)(\sigma_j^2)}}\right]$ | [24] |

Where *IR* represents the pixel value of the infrared band, *R* represents the pixel value of the red-light band. Where *i* represents the gray value of the reference pixel, *j* represents the gray value of adjacent pixels, $P_{i,j}$ is the normalized gray level co-occurrence matrix, which represents the probability that the gray level is the occurrence of a certain relationship between *i* and *j* pixels, $P_{i,i}$ is the normalized gray level co-occurrence matrix, which represents the probability that the gray level is the occurrence of a certain relationship between *i* and *i* pixels, $\mu_i$ represents the mean value, $\sigma_i$, $\sigma_j$ represent the standard deviation of rows and columns respectively, and $\mu_i$, $\mu_j$ represent the mean value of rows and columns respectively.

### 3.2. Classification Algorithms

#### 3.2.1. 3D-CNN

The 2D-CNN only convolves the spatial information, and two-dimensional feature maps are produced regardless of whether the input data is two-dimensional or three-dimensional. Therefore, for the classification of a three-dimensional hyperspectral image, 2D-CNN could lose the most spectral information. The 3D-CNN incorporates both spatial and spectral information, unlike the 2D-CNN. In this way, hyperspectral images can be captured more effectively from a spectral and spatial perspective.

The 3D-CNN is firstly proposed for integrating the spatial and temporal information of video data [31]. As part of a 3D convolution algorithm, a 3D kernel is applied to a cube composed of frames with some columns stacked. The 3D convolution can be expressed as Equation (1).

$$v_{lj}^{xyz} = f\left(\sum_{m}\sum_{h=0}^{H_l-1}\sum_{w=0}^{W_l-1}\sum_{r=0}^{R_l-1} k_{ljm}^{hwr} v_{(l-1)m}^{(x+h)(y+w)(z+r)} + b_{lj}\right) \tag{1}$$

where $H_l$, $W_l$ are the kernel size (height and width), $R_l$ is the size of the 3D kernel along the temporal axis, $l$ identifies the layer considered, $v_{lj}^{xyz}$ stands for the output at position $(x, y, z)$ on the *jth* feature map in the *lth* layer, $b$ is the bias, and $f(\cdot)$ is the activation function, $m$ indexes over the set of feature maps in the $(l-1)$th layer connected to the current feature map, and finally, $k_{ljm}^{hwr}$ contains the $(x, y, z)$ value of the kernel associated with the mth feature cube.

The 3D convolution algorithm applied to HIS classification can be expressed as Equation (2).

$$v_{lji}^{xyz} = f\left(\sum_{h=0}^{H_i-1}\sum_{w=0}^{W_i-1}\sum_{s=0}^{S_i-1} k_{ijm}^{hws} V_{(i-1)j}^{(x+h)(y+w)(z+s)} + b_{ij}\right) \tag{2}$$

A 3D convolution kernel is defined as $S_i$ with $i$ representing the number of feature blocks, and $j$ being the number of convolution kernels. In turn, the three-dimensional feature volume output from the layer *ith* includes $l \times j$.

Different from other 3D-CNN classification models, the proposed model is pixel-level, which is a joint spectral-spatial CNN classification framework. Furthermore, the input data is not the whole image, but the neighbouring spectral-spatial cubes around the pixels. The model, referring to many well-known CNN structures, sets up five convolutional layers, where the next layer contains twice the number of convolutional kernels as the previous one. According to the related study, a $3 \times 3$ convolutional kernel is shown to perform best in spatial feature learning [32]. The convolution kernel is mostly set to $3 \times 3 \times 6$. The processing flow is shown in Figure 3.

1.  Sample phrasing: The original image is cut as a series of spectral-spatial cubes with labels of size S × S × B, and the geometric center pixel of the sample is used as the center. The S × S refers to the size of the neighborhood, while B refers to the number of bands. The cropped S × S × B spectral-spatial cubes are input to the model.

2.  Feature extraction: As shown in Figure 4, there are five convolutional layers and two fully connected layers in the network. First, the S × S × B spectral-spatial cubes are used as input data. In the first convolutional layer (C1), the size of $1 \times N$ convolutional kernels is $K_S^1 \times K_S^1 \times K_D^1$, where N means the number of convolutional kernels in the C1 layer, $K_S^1$, and $K_D^1$ is the mean size of the convolutional kernel in the spatial-spectral dimension. The size of the four-3D cube output in the layer is $\left(S - K_S^1 + 1\right) \times \left(S - K_S^1 + 1\right) \times \left(B - K_D^1 + 1\right)$. After the first convolution layer (C1) is applied, the generated data is applied to the second convolution layer (C2) to get a $2 \times N$ 3D feature cube. The output data of each convolutional layer is the input data of the next convolutional layer. Finally, $16 \times N$ 3D feature cubes with dimensions of $\left(S - K_S^1 - K_S^2 - K_S^3 - K_S^4 - K_S^5 + 5\right) \times \left(S - K_S^1 - K_S^2 - K_S^3 - K_S^4 - K_S^5 + 5\right) \times \left(BS - K_D^1 - K_D^2 - K_D^3 - K_D^4 - K_D^5 + 5\right)$ is output.

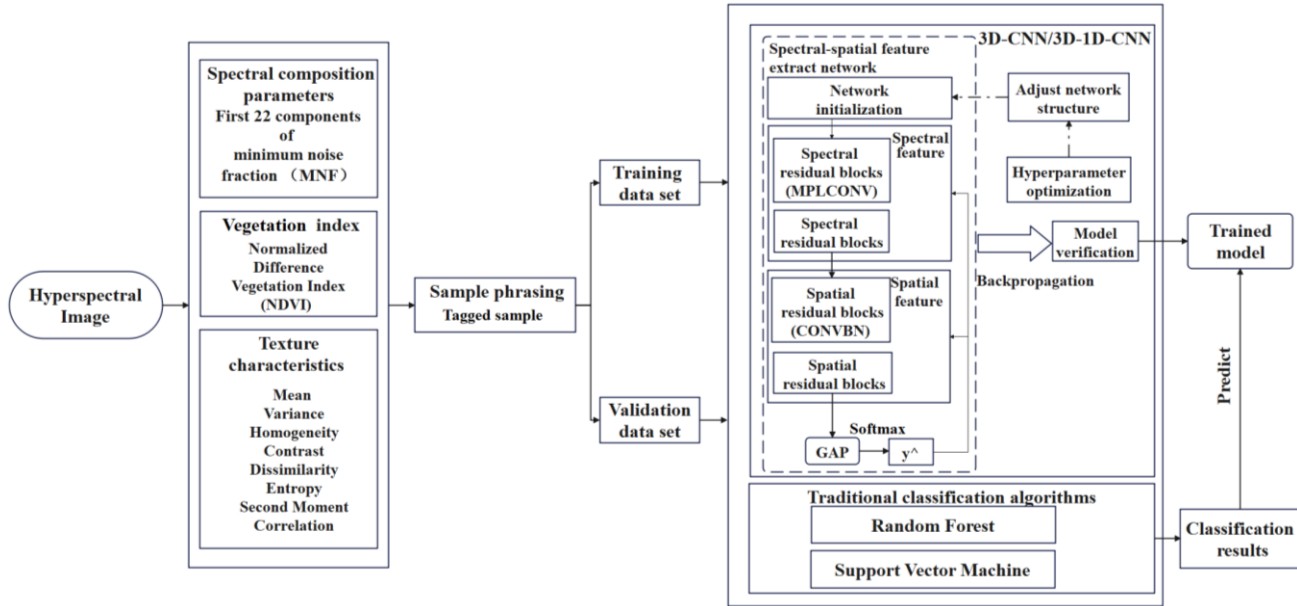

**Figure 3.** The flowchart of this research.

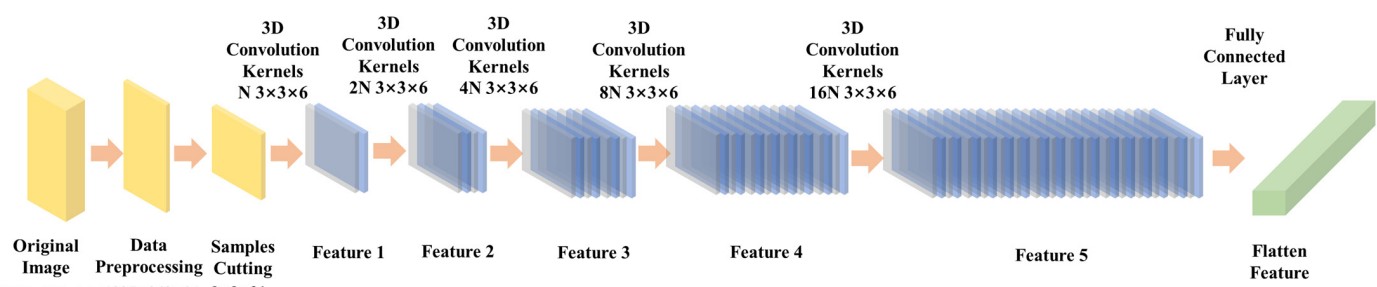

**Figure 4.** 3D-CNN architecture diagram.

3. The last 3D convolutional layer is flattened. The 3D cubes are converted into a $1 \times 128$-dimensional feature vector via the fully connected layer (F1). In the multi-classification task, logistic regression is added behind the fully connected layer. Equation (3) shows the probability that the input characteristics belong to class *i*.

$$P(Y = i \mid V, W, b) = s(WV + b) = \frac{e^{W_i V + b_i}}{\sum_j e^{W_i V + b_i}} \tag{3}$$

In this equation, *W* is the weight, *Y* is the classification result, *b* is the bias, *i* represents the *ith* output unit, and *s* is the SoftMax function.

4. Afterward, the gradient of the backpropagation loss function is used for convolution kernel parameters to be updated.

5. Due to the small sample, stochastic gradient descent is used to complete the process. The linear rectification function [33] is used as the activation function.

6. For overfitting to be avoided, dropout is applied to the model, while the output probability of neurons is set to 0.5 for reducing the interaction of neurons. The learning rate determines when and if the objective function is converged to the local minimum. The epoch is 150. The suitable learning rate can achieve the local minimum during the suitable time. Otherwise, the gradient is discrete or converges slowly. In other words, the learning rate decides the process of each learning iteration. The learning rate of 0.001 is chosen according to the actual test as well as the experience

of scholars [26]. According to the result in the model, the well-performing 3D-CNN structure performance is shown in Table 4.

**Table 4.** 3D-CNN model structure.

| Layer | Output Shape for 3D Data | Training Parameter Number |
|---|---|---|
| conv3d | (9, 9, 26, 4) | 220 |
| conv3d_1 | (7, 7, 21, 8) | 1736 |
| conv3d_2 | (5, 5, 16, 16) | 6928 |
| conv3d_3 | (3, 3, 11, 32) | 27,680 |
| conv3d_4 | (1, 1, 6, 64) | 110,656 |
| dropout | (1, 1, 6, 64) | 0 |
| flatten | (384) | 0 |
| dense | (128) | 49,280 |
| dropout_1 | (128) | 0 |
| dense_1 | (15) | 1935 |

### 3.2.2. Improved 3D-CNN

In attempts to reduce the amount of training parameters and training time, a lightweight 3D-1D-CNN is proposed by Zhang et al. [26]. The 3D-CNN mentioned above is used for feature extraction. To generate a one-dimensional feature representing the high-level semantic concepts captured by the model, a series of 3D cubes are transformed into several 3D convolutional layers. Then, one-dimensional features are used as input for the next stage. Meanwhile, the dense layer is changed in order to represent the features. By reshaping the layers, the 16 N 3D cubes of $(S - K_S^1 - K_S^2 - K_S^3 - K_S^4 - K_S^5 + 5) \times (S - K_S^1 - K_S^2 - K_S^3 - K_S^4 - K_S^5 + 5) \times (BS - K_D^1 - K_D^2 - K_D^3 - K_D^4 - K_D^5 + 5)$ is transformed by five 3D convolution layers into a 16 N dimensional vectors of length $(S - K_S^1 - K_S^2 - K_S^3 - K_S^4 - K_S^5 + 5) \times (S - K_S^1 - K_S^2 - K_S^3 - K_S^4 - K_S^5 + 5) \times (BS - K_D^1 - K_D^2 - K_D^3 - K_D^4 - K_D^5 + 5 + 5)$ and labels its dimension as D. In the first layer of the 1D CNN, H is treated as the same as 6 in the third layer of convolution. The n1 filters are defined to learn a single feature. The input data of the first layer are n1, the one-dimensional vectors $(D - H + 1)$, and one-dimensional convolution is performed by a convolutional kernel of height H and number n2. Then, n2 one-dimensional vectors of length $(D - 2H + 2)$ are gotten. Using a 3D-1D-CNN network structure, Table 5 illustrates the optimal configuration.

**Table 5.** 3D-1D-CNN model structure.

| Layer | Output Shape for 3D Data | Training Parameter Number |
|---|---|---|
| conv3d | (9, 9, 26, 4) | 220 |
| conv3d_1 | (7, 7, 21, 8) | 1736 |
| conv3d_2 | (5, 5, 16, 16) | 6928 |
| conv3d_3 | (3, 3, 11, 32) | 27,680 |
| conv3d_4 | (1, 1, 6, 64) | 110,656 |
| dropout | (1, 1, 6, 64) | 0 |
| Reshape | (6, 64) | 0 |
| conv1d | (4, 48) | 9264 |
| conv1d_1 | (4, 24) | 1176 |
| flatten | (96) | 0 |
| dense | (128) | 12,416 |
| dropout_1 | (128) | 0 |
| dense_1 | (15) | 1935 |

### 3.2.3. Random Forest and Support Vector Machine

Random Forest (RF) [34] is a classifier containing multiple decision trees, and its output categories are determined by the mode of the classes output by individual trees. Unlike CART, the introduction of random features and random data in random forests is

crucial to the classification performance. Figure 5 shows schematic diagram of Random Forest principle.

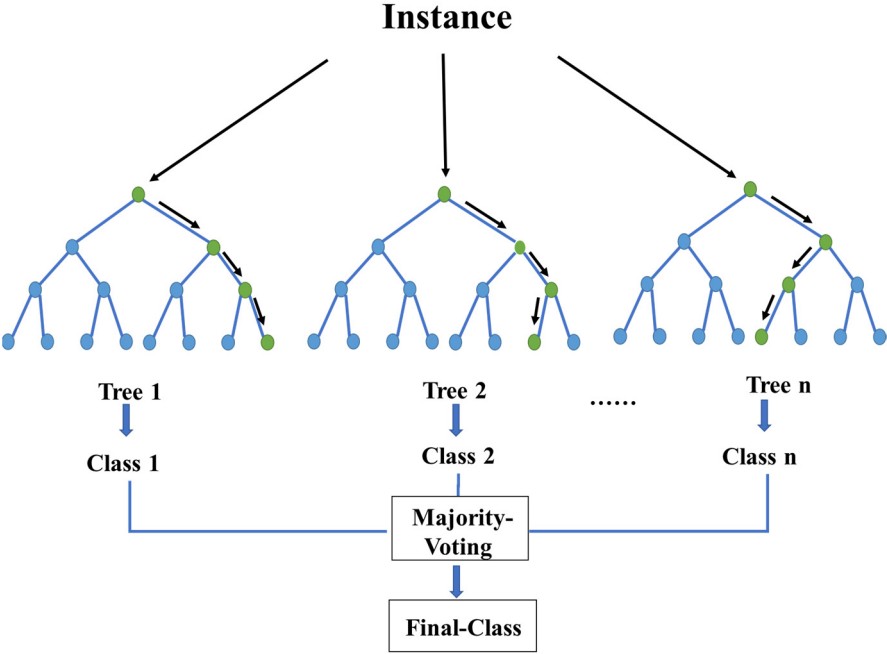

**Figure 5.** Schematic representation of the Random Forest principle.

Support Vector Machine (SVM) is a supervised classification method derived from statistical learning theory that often yields good classification results from complex and noisy data. It separates the classes with a decision surface that maximizes the margin between the classes. The surface is often called the optimal hyperplane, and the data points closest to the hyperplane are called support vectors. The support vectors are the critical elements of the training set. Figure 6 shows schematic diagram of support vector machine.

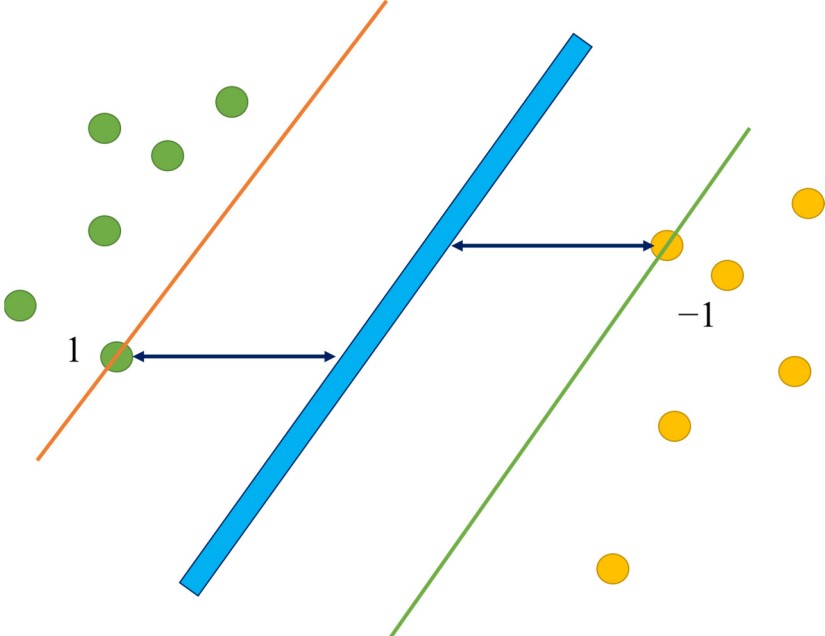

**Figure 6.** Schematic diagram of support vector machine principle.

While SVM is a binary classifier in its simplest form, it can function as a multiclass classifier by combining several binary SVM classifiers (creating a binary classifier for each possible pair of classes). The implementation of SVM uses the pairwise classification strategy [35] for multiclass classification in this paper.

The paired classification method is based on the binary SVM, also called one-against-one [36]. Let the training set data total $M$ classes, one-against-one method is to construct a binary SVM between each two classes. Taking Figure 7a as an example, there are three types of (two-dimensional) data. The dashed line $d_{12}$ represents the decision boundary of binary SVM between Class 1 and Class 2 data; $d_{13}$ represents the decision boundary between Class 1 and Class 3 data; $d_{23}$ represents the decision boundary between Class 2 and Class 3 data. For new data, voting strategy [35] is used for classification. Figure 7b shows the decision boundary drawn according to the voting strategy.

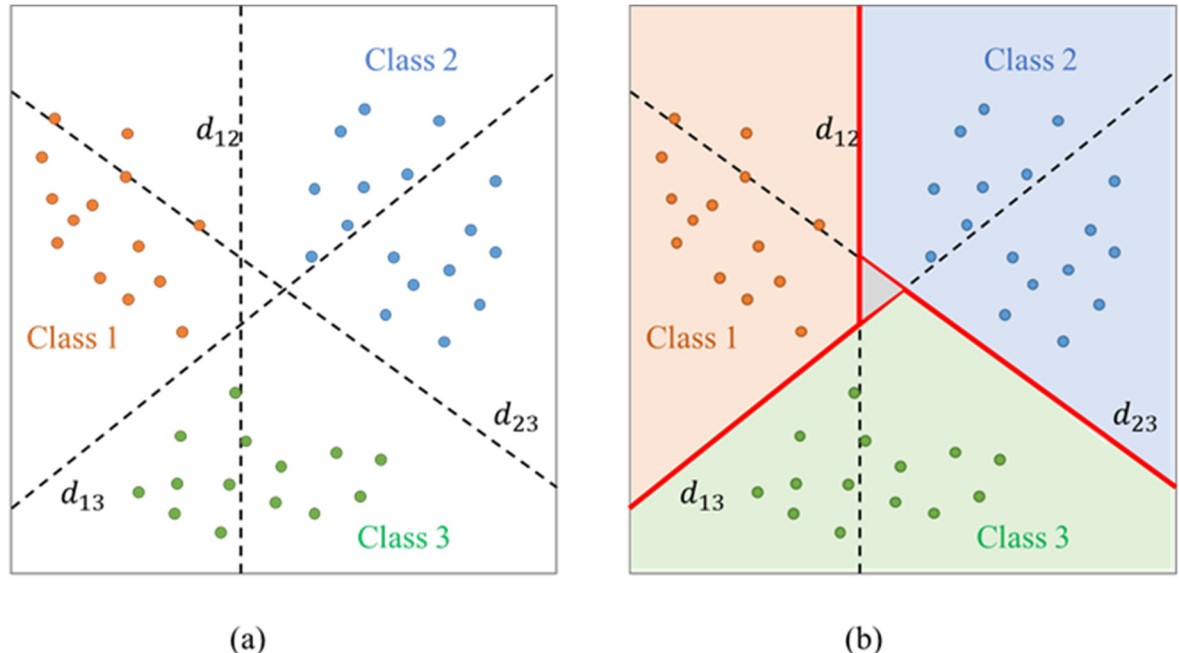

**Figure 7.** The paired classification method. (**a**) Decision boundary, (**b**) Decision boundary drawn according to the voting strategy.

*3.3. Accuracy Assessment*

Using the confusion matrix and the Kappa coefficient (Kappa), how effective the model is in extracting urban fine features can be evaluated. The confusion matrix is calculated by comparing the position and classification of each real image element with the corresponding position and classification in the classified image. The function of the Kappa coefficient is shown in Equation (5):

$$Q = \frac{(TP + FN)(TP + FP) + (TN + FN)(TN + FP)}{TP + TN + FP + FN} \tag{4}$$

$$Kappa = \frac{TP + TN - Q}{TP + TN + FP + FN - Q} \tag{5}$$

In this equation, $TP$ represents the number of classes that are positively predicted, $TN$ represents the number of negatively predicted classes, $FP$ is that wrongly predicts negative classes as positive, $FN$ indicates how many positive classes are predicted to be negative, and $Q$ represents the intermediate variable in the *Kappa* calculation.

Overall accuracy ($OA$) is used to assess the overall quality of the results. When the values of these indicators are high, then the predicted results and actual results are more in agreement. The calculation of these indicators is shown in Equation (6):

$$OA = \frac{TP + TN}{TP + TN + FP + FN} \tag{6}$$

Evaluation of the network performance is based on Precision ($P$), recall ($R$), and F1-score ($F1$). In feature extraction tasks, the higher the precision, recall, and F1-score, the lower the false detections and omission of the changed pixels.

$$P = \frac{TP}{TP + FP} \tag{7}$$

$$R = \frac{TP}{TP + FN} \tag{8}$$

$$F_1 = \frac{2PR}{P + R} \tag{9}$$

## 4. Results

The Support Vector Machine (SVM) and Random Forest (RF) are also employed for comparison. The 3D-CNN and 3D-1D-CNN are programmed in Pycharm and Jupyter Notebook of Anaconda3, based on the TensorFlow and Keras frameworks. AMD Ryzen 7 5800H with Radeon Graphics @ 3.20 GHz CPU and NVIDIA GeForce GTX 3060 GPU comprise the operating platform hardware configuration.

### 4.1. Houston Dataset

Compared with 3D-CNN, the overall accuracy (OA) of 3D-1D-CNN is improved by 0.42% (from 95.90% to 96.32%). Compared with 3D-CNN and 3D-2D-CNN, the training parameters of 3D-1D-CNN are shortened by 13.32% (from 198,435 to 172,707) and 71.46% (from 602,743 to 172,707), the training time is shortened by 37.95% (from 5.77 min to 3.58 min) and 54.91% (from 7.94 min to 3.58 min). Table 6 and Figure 8 also indicate that the results of 3D-CNN and 3D-1D-CNN are better than that of the SVM and RF classifiers.

**Table 6.** Performance comparison of different classification methods on the Houston dataset.

|  | SVM | RF | 1D-CNN | 3D-CNN | 3D-2D-CNN | 3D-1D-CNN |
|---|---|---|---|---|---|---|
| commercial | 0.56 | 0.72 | 0.98 | 1.00 | 1.00 | 0.99 |
| grass_healthy | 0.88 | 0.88 | 0.90 | 0.93 | 0.93 | 0.88 |
| grass_stressed | 0.89 | 0.96 | 0.98 | 1.00 | 0.99 | 1.00 |
| grass_synthetic | 0.92 | 1.00 | 0.57 | 1.00 | 1.00 | 1.00 |
| highway | 0.56 | 0.73 | 0.91 | 1.00 | 1.00 | 1.00 |
| parking_lot1 | 0.47 | 0.84 | 0.67 | 1.00 | 0.83 | 1.00 |
| parking_lot2 | 0.70 | 0.81 | 0.83 | 1.00 | 1.00 | 0.97 |
| railway | 0.59 | 0.80 | 0.54 | 0.42 | 0.62 | 0.44 |
| residential | 0.63 | 0.77 | 0.94 | 1.00 | 0.98 | 1.00 |
| road | 0.58 | 0.82 | 0.90 | 0.98 | 0.98 | 1.00 |
| running_track | 0.98 | 0.98 | 0.95 | 1.00 | 1.00 | 1.00 |
| soil | 0.97 | 0.97 | 0.99 | 1.00 | 1.00 | 1.00 |
| tennis_court | 0.93 | 0.94 | 0.83 | 0.75 | 1.00 | 1.00 |
| tree | 0.93 | 0.90 | 0.99 | 0.96 | 0.98 | 0.98 |
| water | 0.95 | 0.97 | 0.92 | 1.00 | 0.96 | 1.00 |
| AA (%) | 77.71 | 80.62 | 85.67 | 93.64 | 95.15 | 94.97 |
| OA (%) | 72.36 | 85.30 | 91.10 | 95.90 | 96.32 | 96.32 |
| Kappa $\times$ 100 | 70.01 | 84.04 | 90.08 | 95.42 | 95.89 | 95.89 |
| Training parameters | - | - | - | 198,435 | 602,743 | 172,011 |
| Training time(minute) | - | - | - | 5.77 | 7.94 | 3.58 |

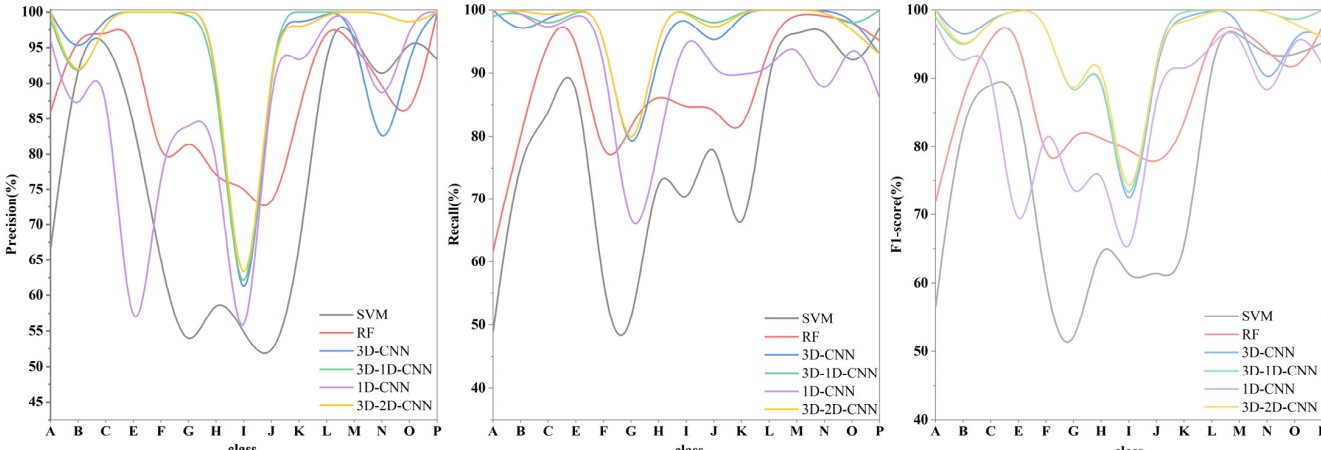

**Figure 8.** Recall, Precision, and F1-score of each class in different classifiers on the Houston dataset, where A, B, C, D, E, F, G, H, I, J, K, L, M, N, O, represent commercial, grass_healthy, grass_stressed, grass_synthetic, highway, parking_lot1, parking_lot2, railway, residential, road, running_track, soil, tennis_court, tree, water, respectively.

Figure 8 shows that the performance of SVM and RF is relatively low in precision (below 80%) in the following four classes: highway, parking_lot2, railway, and residential, and the performance of SVM and RF is relatively low in the recall (below 79%) in the following four classes: highway, commercial, road. The performance of 1D-CNN in grass_synthetic, parking_lot2, and railway is poor. However, the precision and recall of the fifteen classes in 3D-CNN and 3D-1D-CNN are rarely below 80%. The results of 3D-2D-CNN and 3D-1D-CNN are similar. Therefore, they have higher classification accuracy than SVM and RF.

In 3D-CNN, the precision of the railway is extremely low, which means that other classes are misclassified as a railway. The 3D-1D-CNN is confused for recognition on parking_lot1 and railway. As Figure 9 shows, spectral similarity may be the main reason for misclassification. Meanwhile, the possibility of commission and omission errors in 3D-1D-CNN is lower than that of 3D-CNN.

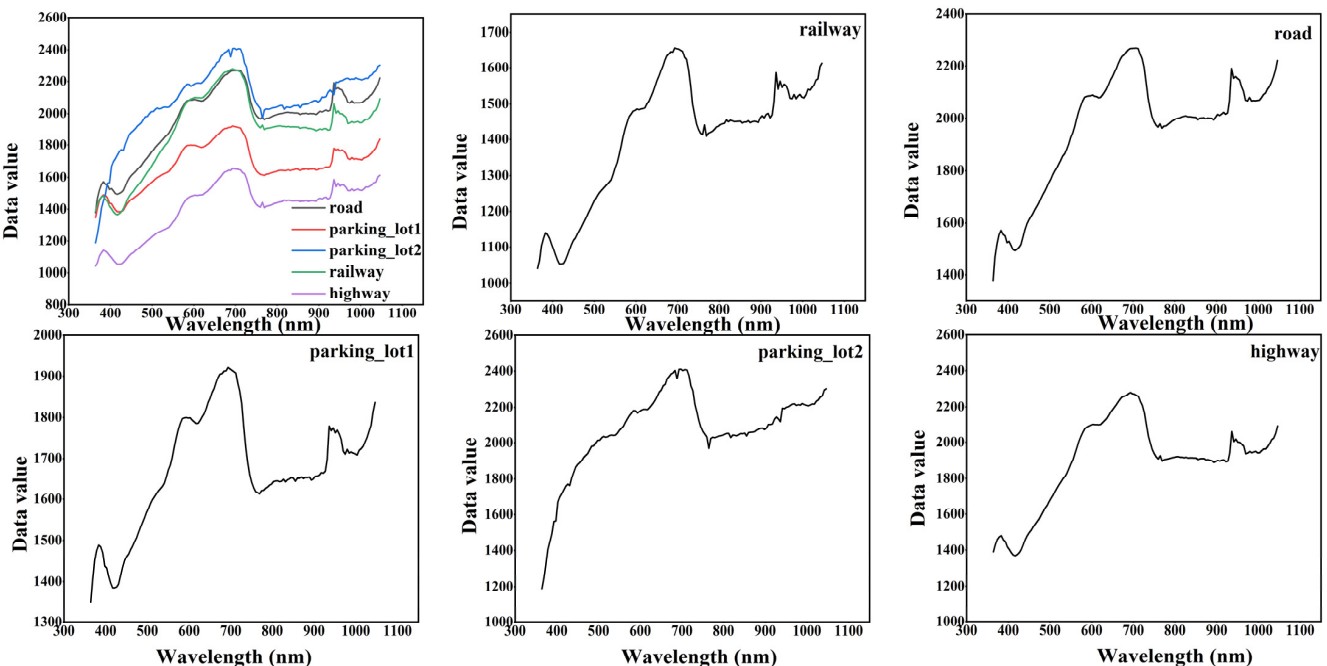

**Figure 9.** Spectral curves of classes with commission and omission error on Houston dataset.

Due to the loss of spectra, urban feature extraction is extremely challenging in cloud shadow areas. Figure 10 shows the difference in the spectral curve in the shadow and non-shadow areas. Furthermore, feature extraction is limited by various building types, irregular boundaries, and spectral similarities. Therefore, the classification results of the traditional classifiers (SVM and RF) are not ideal in the cloud shadow area. However, the results of 3D-CNN, 3D-2D-CNN, and 3D-1D-CNN in cloud shadow areas are much superior to that of SVM and RF classifiers (Figure 11), especially in the following classes: residential, commercial, grass_synthetic, grass_stressed, water, tennis_court, road, soil, and grass_healthy. It can be concluded that even if the spectrum is missing in the cloud shadow area, the 3D-1D-CNN can still have a good performance in training and predicting.

Networks with fewer convolutional layers usually have lower semantics and more noise due to less convolutional processing. The target method sets five 3D convolution layers and two 1D convolution layers. Through multiple convolutional processing, the variation of deep features in the image becomes larger, the semantics is improved, and the noise is reduced, to highlight the discrimination between images. In addition to the above reasons, the features of the ground objects cannot be ignored. NDVI has a high sensitivity for vegetation detection. It is a comprehensive reflection of vegetation types, cover forms, and growth conditions in a unit pixel, and is often applied to the monitoring of surface vegetation. In this study, NDVI, as part of the input data, highlights the distinction between vegetation and between vegetation and non-vegetation. This is also a reason why the target method is better for vegetation classification. Parking_lot1 is a parking lot with no cars, and parking_lot2 is a parking lot with cars. The presence of cars intensifies the reflection of the sun's rays, making the reflectivity of parking_lot2 different from that of parking_lot1 (Figure 9). In addition, the presence of cars distinguishes parking_lot2 from other ground objects (such as railway) that are composed of concrete, gravel, and other components. Parking_lot1 is misclassified as railway, since both have the same composition and both have no vehicle distribution. These may be the reasons for the good performance of 3D-1D-CNN in parking_lot2, grass_healthy, grass_stressed, grass_synthetic extraction.

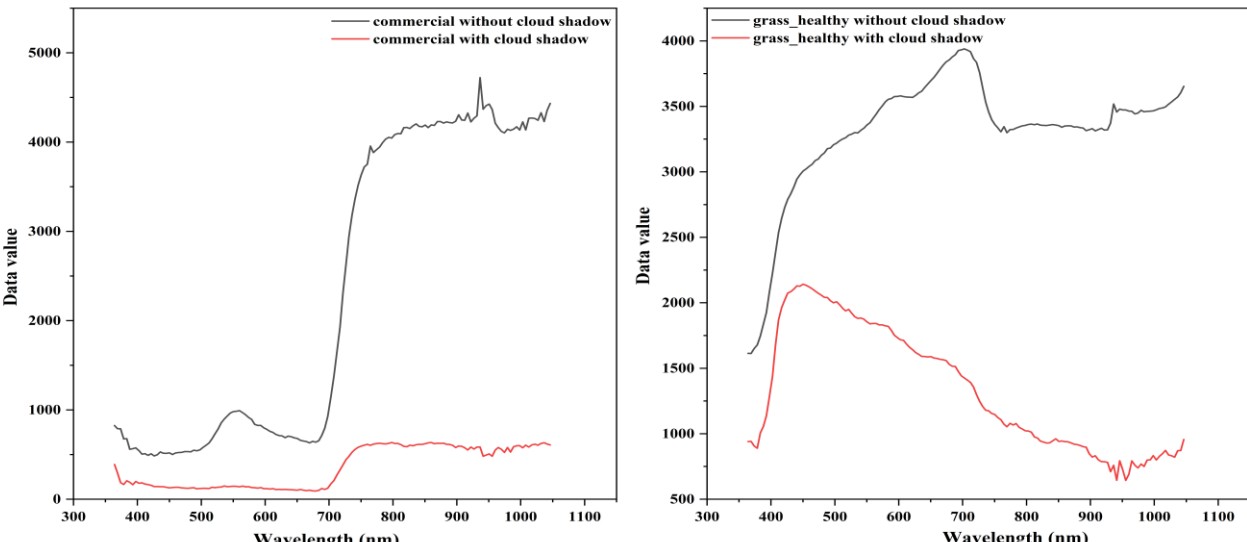

**Figure 10.** Comparison of spectral curves of the same class on Houston dataset. The (**left**) is the spectral curves of *commercial*, the (**right**) is the spectral curves of *grass_healthy*.

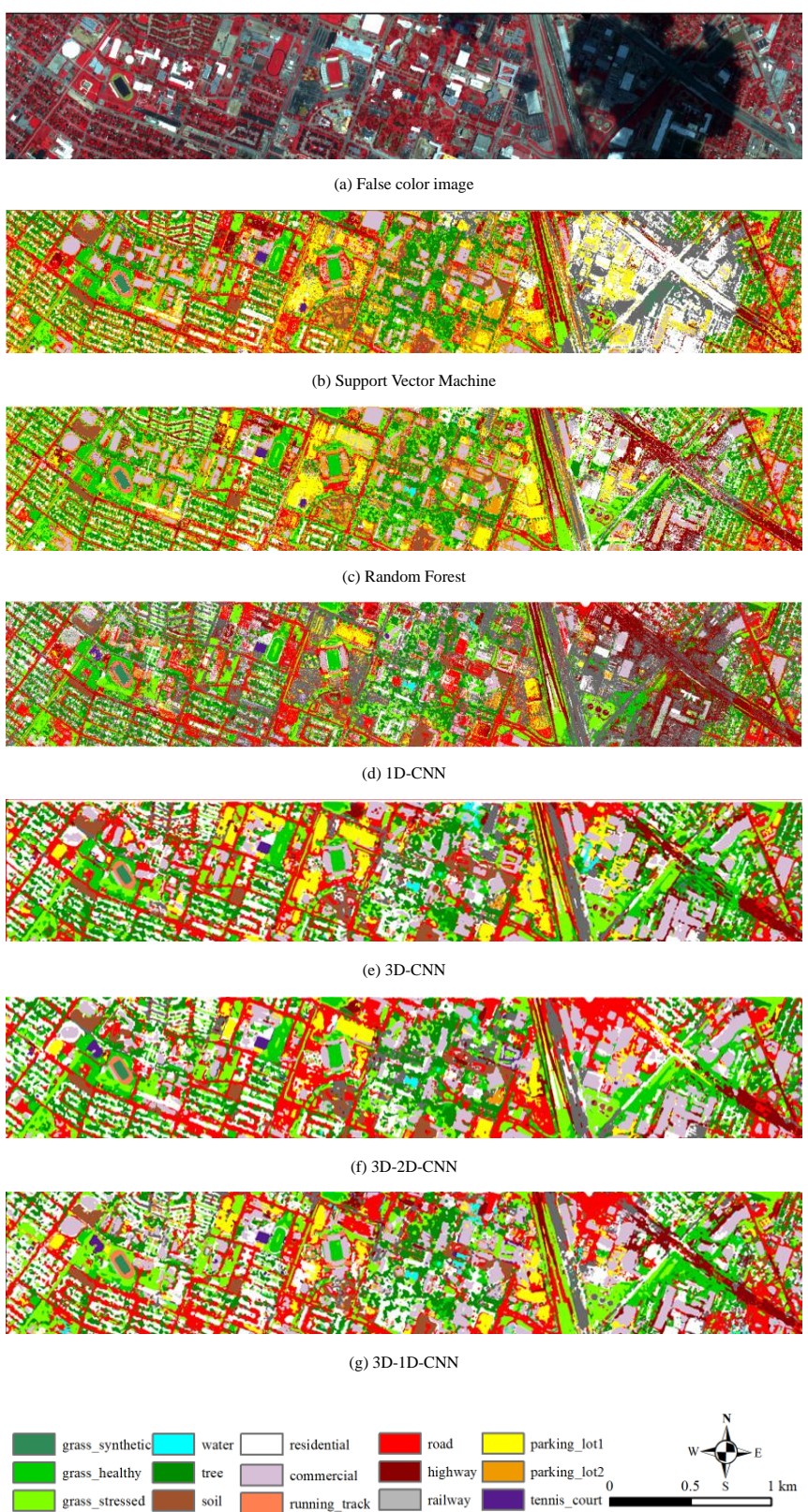

**Figure 11.** Classification results of different methods for Houston dataset with 15 classes. (**a**) False color image. (**b**) The classification map of SVM classifier, OA = 72.36%. (**c**) The classification map of RF classifier, OA = 85.30%. (**d**) The classification map of 1D-CNN classifier, OA = 91.10%. (**e**) The classification map of 3D-CNN classifier, OA = 95.90%. (**f**) The classification map of 3D-2D-CNN classifier, OA = 96.32%. (**g**) The classification map of 3D-1D-CNN classifier, OA = 96.32%.

### 4.2. Surrey Dataset

Compared with SVM and RF classifier, the overall accuracy (OA) of 3D-1D-CNN is increased by 7.63% and 3.61%, respectively. Compared with 3D-CNN, the training parameters of 3D-1D-CNN are shortened by 13.27% (from 199,131 to 172,707), the training time is shortened by 6.89% (from 10.16 min to 9.46 min). Table 7 also shows that the results of 3D-CNN and 3D-1D-CNN are better than those of SVM and RF classifiers. As Zhang et al. [26] said, 3D-1D-CNN reduces the training time and training parameters while maintaining a high accuracy. It can be seen from Table 7 that 3D-1D-CNN has high classification accuracy on the four classes of tree, grass, soil and water, and low accuracy (less than 80%) in the parking_lot.

**Table 7.** Accuracy of each class with different classification methods for Surrey dataset.

| | SVM | RF | 1D-CNN | 3D-CNN | 3D-2D-CNN | 3D-1D-CNN |
|---|---|---|---|---|---|---|
| tree | 0.94 | 0.94 | 0.85 | 0.95 | 0.91 | 1.00 |
| grass | 1.00 | 0.99 | 0.97 | 0.76 | 0.97 | 1.00 |
| soil | 0.92 | 0.80 | 0.50 | 0.29 | 0.60 | 1.00 |
| water | 0.96 | 0.98 | 0.90 | 0.78 | 0.86 | 1.00 |
| parking_lot | 0.75 | 0.86 | 0.80 | 0.58 | 0.78 | 0.75 |
| road | 0.66 | 0.74 | 0.93 | 0.97 | 0.94 | 0.82 |
| building | 0.81 | 0.90 | 1.00 | 1.00 | 0.92 | 0.88 |
| OA (%) | 82.81 | 86.83 | 88.20 | 82.58 | 89.33 | 90.44 |
| Kappa × 100 | 78.96 | 83.78 | 84.95 | 78.05 | 86.29 | 87.72 |
| Training parameters | - | - | - | 199,131 | 601,711 | 172,707 |
| Training time (minute) | - | - | - | 10.16 | 11.30 | 9.46 |

Comparison of the spectral reflection curves of the classes with lower classification accuracy is shown in Figure 12. The spectral reflection curves of road, parking _lot and building are very similar, which may be the main reason for the lower classification accuracy. Additionally, the presence of cars in the parking_lot also affects the correct identification of the parking_lot by 3D-1D-CNN. However, the accuracy of tree and grass is relatively high with 3D-1D-CNN. The use of vegetation index might be the reason.

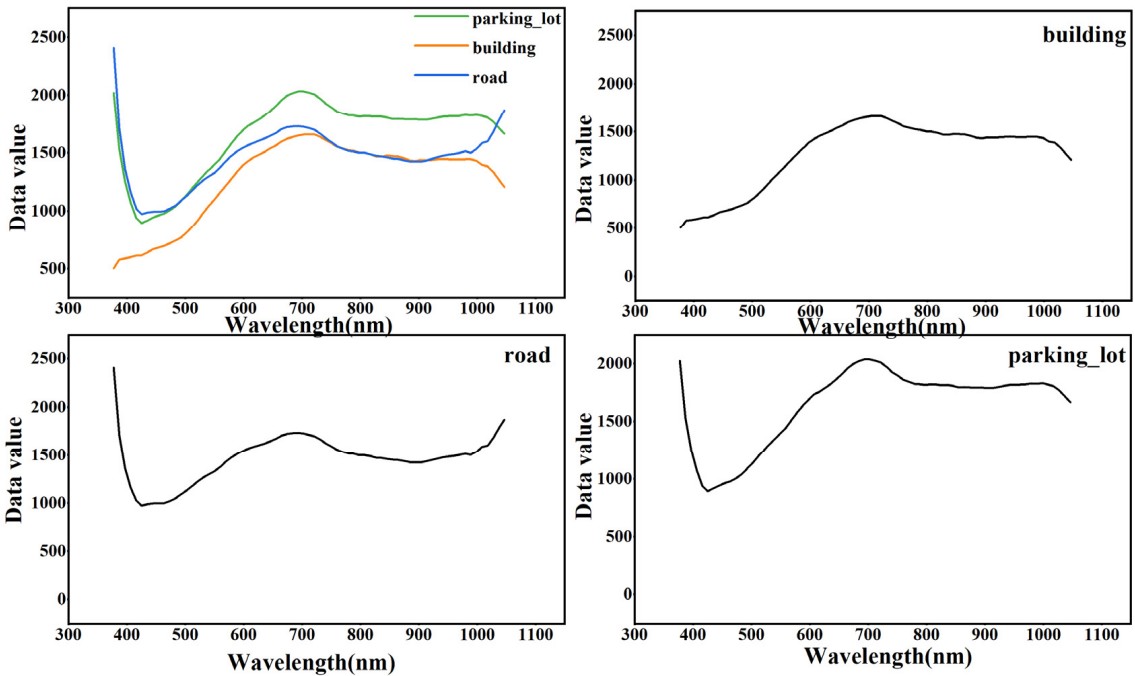

**Figure 12.** Spectral curves of classes with commission and omission error on Surrey dataset.

## 5. Discussion

### 5.1. Impact of Hyperspectral Datasets on 3D-1D-CNN

Since this paper is mainly based on the Houston dataset, which is a public popular dataset, in order to verify the transferability of 3D-1D-CNN, we choose another study area (Surrey dataset) to verify the performance of the target model. The new study area is in city of Surrey.

The overall accuracy of Surrey dataset is slightly worse than the overall accuracy of the Houston dataset. The difference may be caused by different spatial spectral resolution or different seasons [37]. Yan, Y. et al. [38] improved the spectral resolution of hyperspectral images to improve the classification accuracy of convolutional neural networks for large hyperspectral datasets. Therefore, spectral resolution plays an important role in the classification performance of convolutional neural networks. The different resolutions as well as the seasons also validate the transferability of the target method. In the near future, the effects of season and resolution on the classification of 3D-1D-CNN models will be discussed. In a nutshell, 3D-1D-CNN has certain transferability.

### 5.2. Impact of Hyperspectral Parameters on 3D-1D-CNN for Houston Dataset

To investigate the effect of hyperspectral parameters on the model, we compared the model results for different combinations of spectral composition parameters, vegetation index and texture characteristics used in this paper. Adding texture characteristics is not as effective as adding vegetation index. This phenomenon may occur because after principal component analysis, for texture characteristics, they only contain less information about the original data. As Table 8 shows, the combination of spectral composition parameters with either the texture characteristics or vegetation index gives good classification results. The addition of vegetation index also improves the effect of using spectral composition parameters alone. Li et al. [39] demonstrated the combination of vegetation index as well as texture characteristics to improve the accuracy of tree species classification in clouds. Therefore, the combinations of spectral composition parameters, texture characteristics and vegetation index can improve the ability of 3D-1D-CNN to mine spatial-spectral information in hyperspectral data.

**Table 8.** Comparison of the performance of different hyperspectral parameters on Houston dataset (input size = 9).

|  | MNF22 | MNF22 + NDVI | MNF22 + GLCM | MNF22 + GLCM + NDVI |
|---|---|---|---|---|
| commercial | 1.00 | 1.00 | 0.99 | 0.99 |
| grass_healthy | 0.89 | 0.85 | 0.79 | 0.88 |
| grass_stressed | 0.97 | 1.00 | 0.99 | 1.00 |
| grass_synthetic | 1.00 | 1.00 | 0.67 | 1.00 |
| highway | 0.96 | 0.96 | 0.91 | 1.00 |
| parking_lot1 | 0.95 | 0.86 | 0.88 | 1.00 |
| parking_lot2 | 1.00 | 1.00 | 1.00 | 0.97 |
| railway | 0.44 | 0.41 | 0.44 | 0.44 |
| residential | 1.00 | 1.00 | 0.97 | 1.00 |
| road | 0.98 | 0.99 | 0.94 | 1.00 |
| running track | 1.00 | 1.00 | 1.00 | 1.00 |
| soil | 1.00 | 1.00 | 1.00 | 1.00 |
| tennis court | 0.75 | 0.86 | 0.86 | 1.00 |
| tree | 0.93 | 0.93 | 0.98 | 0.98 |
| water | 1.00 | 1.00 | 1.00 | 1.00 |
| AA (%) | 92.44 | 92.39 | 89.33 | 94.97 |
| OA (%) | 94.48 | 93.92 | 93.07 | 96.32 |
| Kappa × 100 | 93.84 | 93.22 | 92.27 | 95.89 |

### 5.3. Impact of Different Input Sizes (Window Sizes) on 3D-1D-CNN for Houston Dataset

In this paper, training samples with different input sizes ($7 \times 7$, $9 \times 9$, $11 \times 11$, $13 \times 13$, and $15 \times 15$) were input into 3D-1D-CNN to analyse the effects of different input sizes on classification accuracy. The results of the classifications are shown in Figure 13.

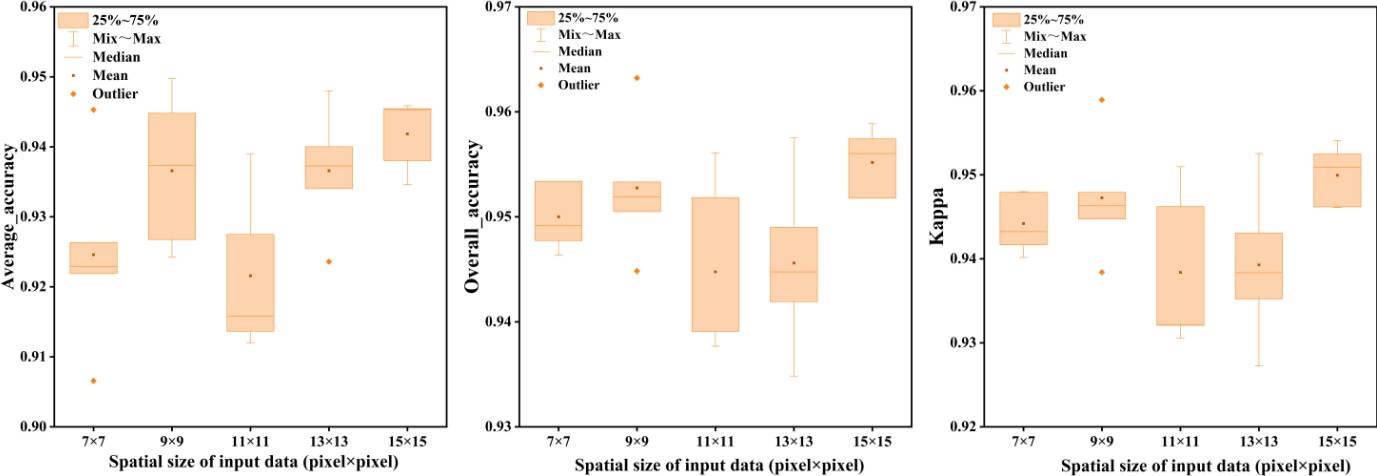

**Figure 13.** Impact of different input sizes on 3D-1D-CNN for Houston dataset.

The results show that the overall trend of the mean value of accuracy is upward with the increasing spatial size of the input data (pixel $\times$ pixel). Each time the parameters and weights of the neural network are random. When training a neural network, it may search for the local optimal solution instead of the global optimal solution. Because of this, even when the center pixel and its neighboring pixels belong to the same class, the accuracy of the $11 \times 11$ input size may suddenly decrease. Usually, the spectral-spatial features extracted from the neighborhood pixels could help reduce the intraclass variance and improve the classification performance. However, the larger the input size, the more noise it may contain, especially the pixels which are located at the corner or edge of a class (Figure 14). Therefore, the Kappa and overall accuracy of $15 \times 15$ input size are uneven. Through many experiments, an input size of $9 \times 9$ and $15 \times 15$ seems to be the best window size in spectral-spatial extraction. Figure 13 demonstrates that the mean of OA, AA, and Kappa for the $15 \times 15$ input size are all higher than the other input sizes. However, the best results of several experiments appeared in an input size of $9 \times 9$, and the training time under this input size is the shortest. Therefore, the input size of $9 \times 9$ is chosen for the final experiment in this paper.

### 5.4. Impact of 3D-1D-CNN on Cloud Shadows for Houston Dataset

Aiming for exploring the impact of 3D-1D-CNN in the cloud shadow area, the results of RF, SVM, 1D-CNN, 3D-CNN, 3D-2D-CNN, and 3D-1D-CNN in the cloud shadow region were intercepted. The hyperspectral parameters used for all classifiers were MNF22+GLCM+NDVI. In addition, the results of different classifiers in cloud shadow areas are shown in Figure 15.

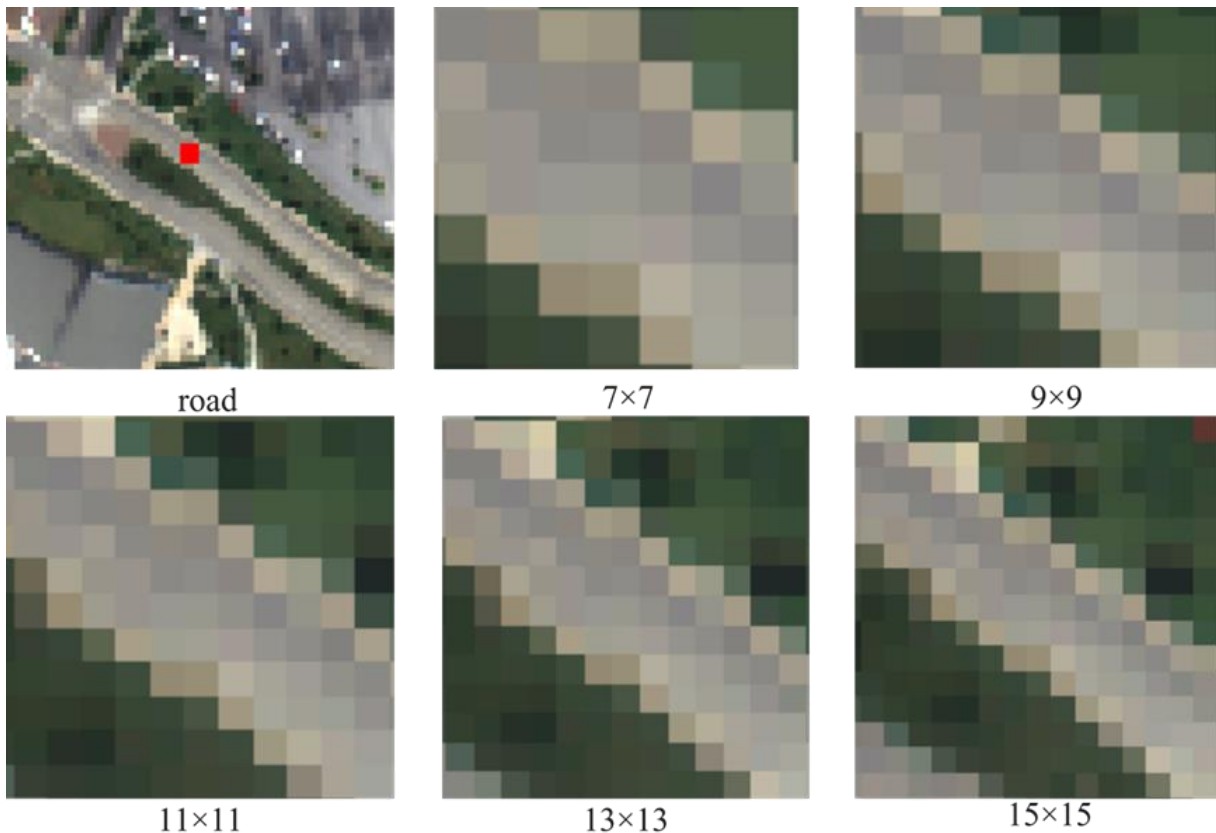

**Figure 14.** Sample of road with different input sizes on the Houston dataset.

As Figure 15 shows, the RF classifier and 1D-CNN could extract buildings (commercial and residential) accurately. Whereas the SVM classifier hardly extracts urban features with cloud shadow areas. This result is consistent with the results of Man et al. [19]. However, these three methods that only use spectral information have the salt and pepper effect. On the contrary, 3D-CNN, 3D-2D-CNN, and 3D-1D-CNN perform well in this regard, especially in the following classes, such as grass, tree, highway, commercial and residential. The 3D-1D-CNN is the best algorithm among the four algorithms to identify highways in cloud shadow areas. There are two reasons why the cloud shadow region restricts the feature extraction of the classifier. One is the loss of some spectra, and the other one is spectral similarity (Figure 16). Under such circumstances, the result of 3D-1D-CNN is much better than that of traditional classification methods in spectral-spatial information mining in the cloud shadow area. However, the resulting graph is not continuous and regular. This problem has also arisen in other studies [40]. Li et al. [40] believed that the dimensionality reduction process resulted in the loss of the external contour of classes. A large number of continuous regular training samples will also affect the model's ability to identify classes. Up to now, there has been less research on urban feature extraction in cloud shadow areas using 3D-1D-CNN.

To exclude the effect of atmospheric correction on the recognition ability of features in cloud-shaded areas, the classification effects of different methods in the cloud shadow regions with and without atmospheric correction are compared in Figure 16. The 3D-1D-CNN also has good recognition ability for classes in cloud-shaded areas without atmospheric correction. Therefore, the perfect recognition ability for classes in cloud shadow areas comes from the 3D-1D-CNN model itself.

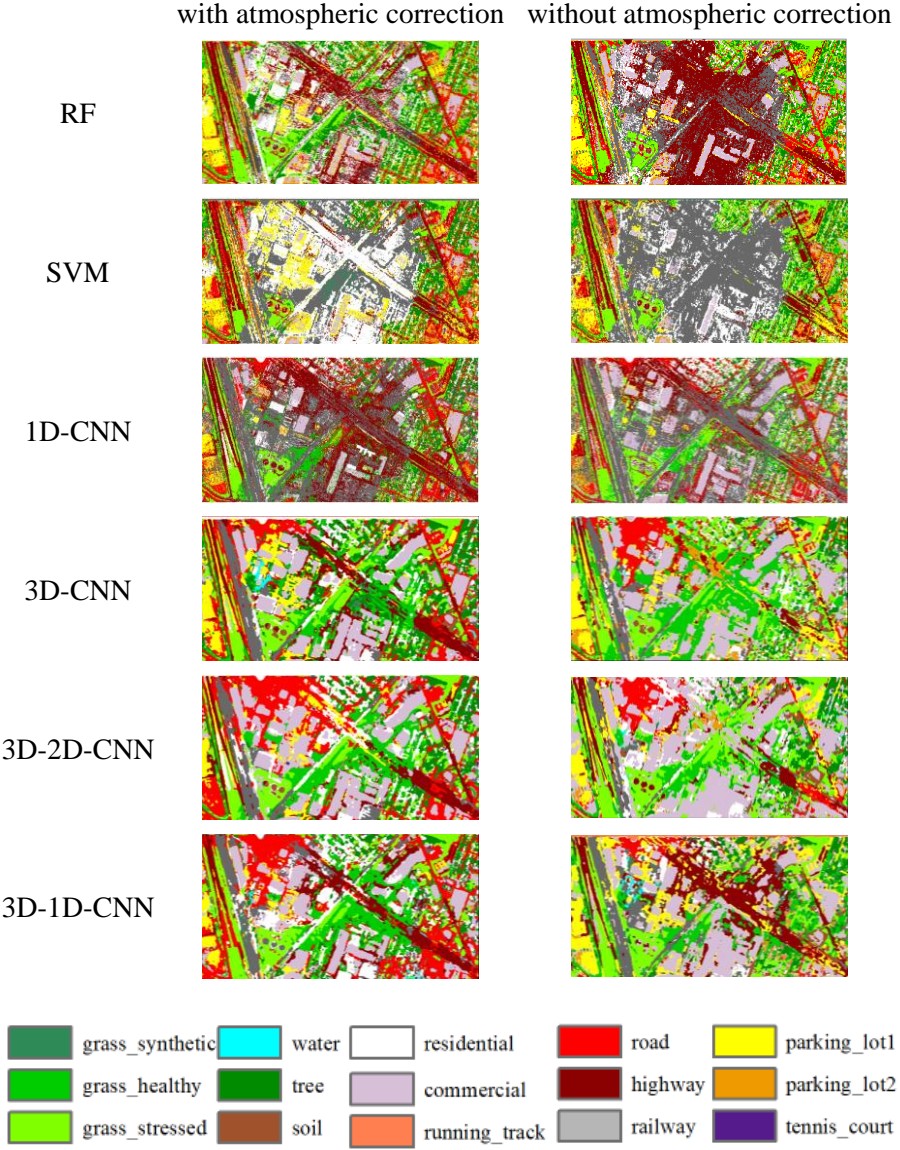

**Figure 15.** Classification results in the cloud shadow area of the Houston dataset.

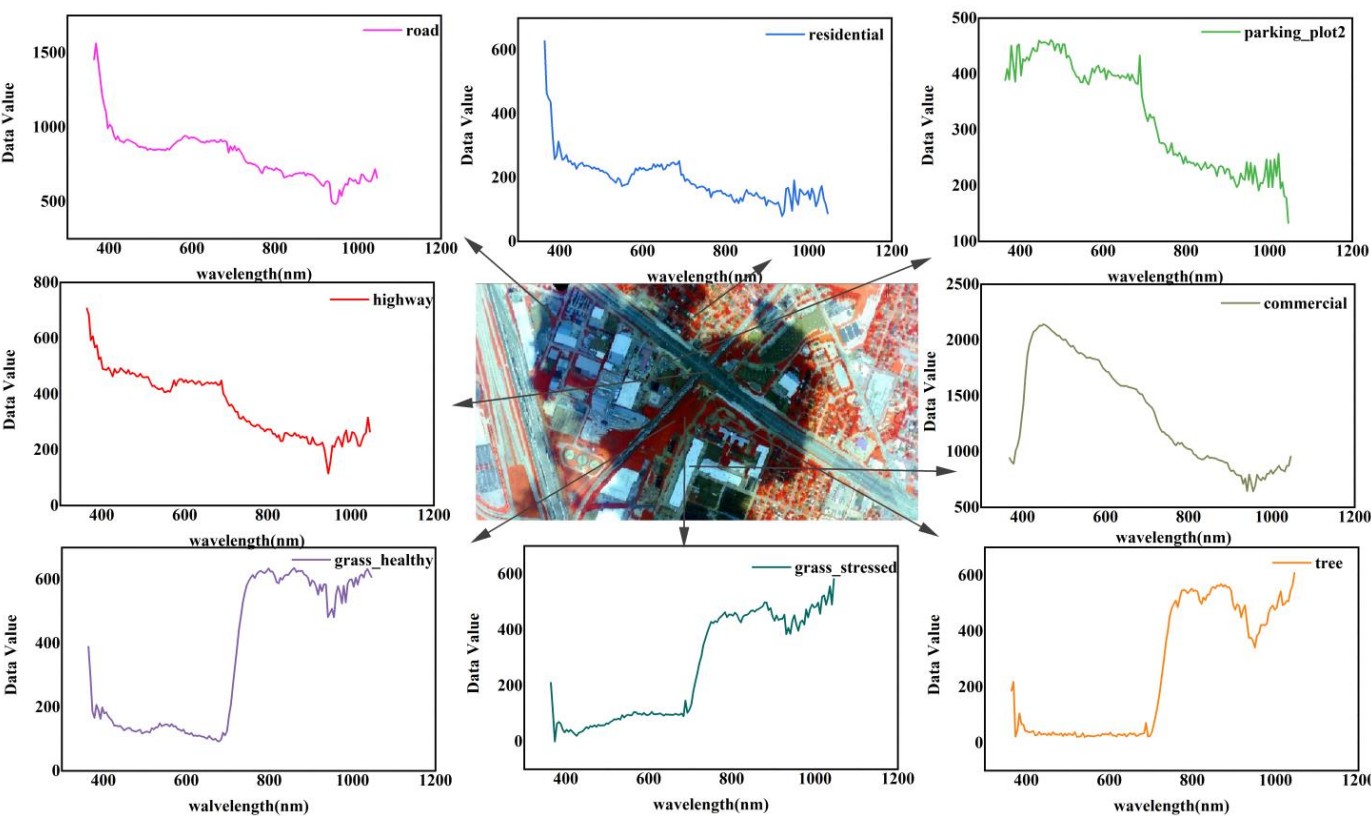

**Figure 16.** Spectral curves of classes in the cloud shadow area.

### 5.5. Impact of 3D-1D-CNN on the Mining Hyperspectral Spatial-Spectral Information

Aiming for exploring the performance of 3D-1D-CNN on the mining hyperspectral spatial-spectral information, the performance of different deep learning models on the Houston dataset were compared. The confusion matrix, loss and accuracy curves of 3D-CNN, and 3D-1D-CNN in the Houston dataset were compared (Figures 17 and 18).

As Figure 17 shows, the number of true positives of parking_lot2, grass_healthy, and residential in 3D-1D-CNN is more than that in 3D-CNN. It means that 3D-1D-CNN is much more effective in hyperspectral classifications. However, whether 3D-CNN or 3D-1D-CNN, the classification result between parking_lot1 and railway is still terrible. The reason is that parking_lot1 and railway have similar spectral characteristics. Therefore, parking_lot1 is misclassified as a railway in the models. As Figure 18 shows, the convergence time of the accuracy curve is roughly the same, which is 50 epochs. In addition, increasing the number of epochs can slightly improve accuracy. After 150 epochs, all curves converge of 3D-1D-CNN. The slight fluctuation of the loss function in the validation dataset is due to the small number of training samples. Compared with 3D-CNN, the overall accuracy (OA) of 3D-1D-CNN is improved by 0.42% (from 95.90% to 96.32%). Table 6 and Figure 8 also indicate that the results of 3D-CNN and 3D-1D-CNN are better than that of the SVM and RF classifiers. It can be seen from Table 6 that 3D-1D-CNN uses fewer training parameters to achieve higher accuracy in a short time. Therefore, 3D-1D-CNN model can well extract the features of classes in the complex urban area. As Zhang et al. [26] mentioned that 3D-1D-CNN requires fewer training parameters and time, with lower misclassification probability, compared with 3D-CNN. Although 3D-1D-CNN simplifies the structure of 3D-CNN, it is still good at mining spatial-spectral information of classes.

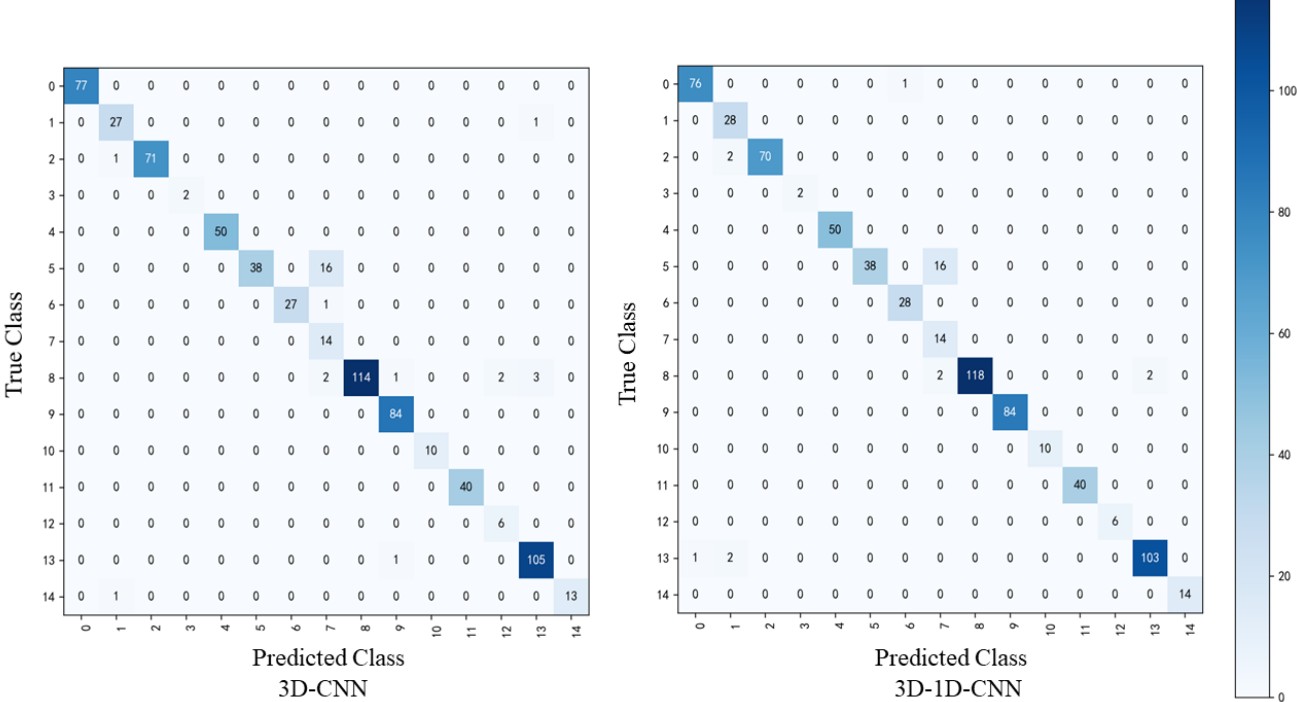

**Figure 17.** Confusion matrix comparison of different models on the Houston dataset.

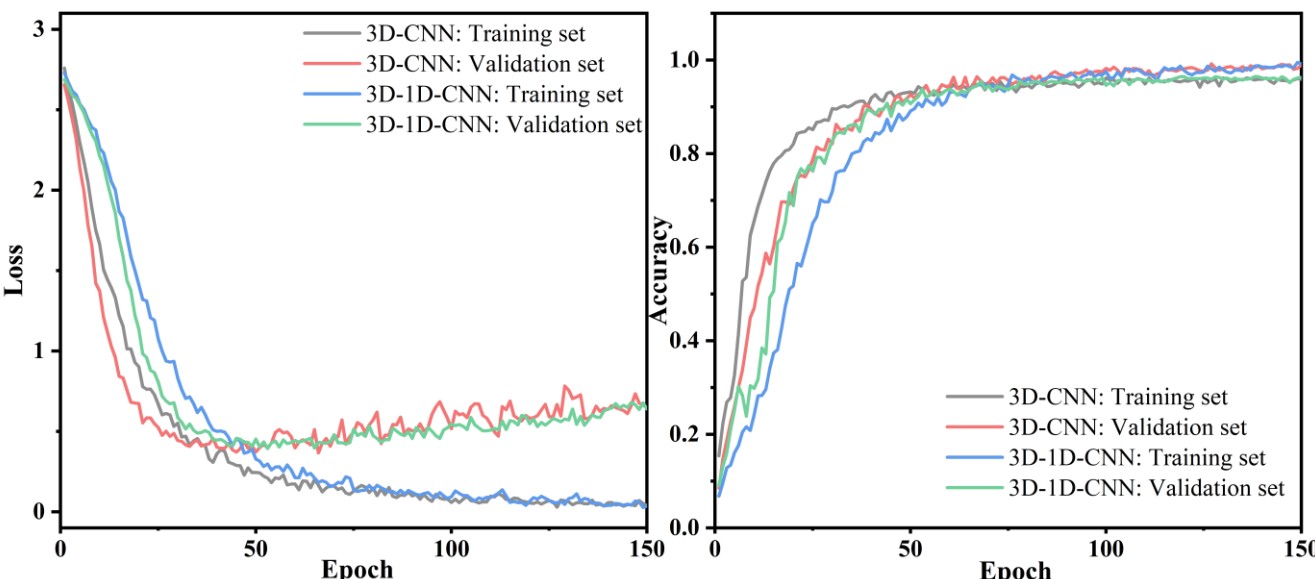

**Figure 18.** Comparison of accuracy loss curves of different models on the Houston dataset.

In addition, examples of classification performed by different deep learning models on the Houston dataset are compared. Zhang et al. [41] achieved 88.80% overall accuracy using 2832 training samples, which is 7.52% less than our results. Feng et al. [42] achieved 96.31% of overall accuracy, which is 0.01% less than our results. The samples used in other studies are all the pixels of the region of interest (ROI), while the samples used in this paper are the central pixel and a few random pixels within the shape of ROI. Therefore, the samples used in other studies contain more pixels, which makes classification easier. It is widely known that if the used training images are different, it is meaningless to compare the overall accuracy across two different experiments because they are not calculated against the same benchmark. Therefore, Li et al. [40] achievement of 97.22% on 15030 samples does

not mean the method can achieve 97.22% on this image. Shi et al. [43] proposed 3DCAM, which achieved 97.69% of overall accuracy. Because there are many other 3D-CNN-derived models out there and still being developed, it is good enough to use RF, SVM, 1D-CNN, 3D-CNN, and 3D-2D-CNN as benchmarks. It is impossible to keep up with all of them.

We depend on the accurately manually labelled samples to calculate the error matrix accuracy as the globally-comparable metrics to evaluate the effectiveness of this study compared to other recent research. However, even though the other papers are using the same dataset (e.g., the Houston dataset), due to the differences in the spatial/spectral pattern and feature distribution of the chosen areas and image subsets, the reported higher overall accuracy in other papers cannot guarantee their approaches will still hold the advance in this use case. One of the major contributions of this paper is providing another data point on the performance of 3D-1D-CNN in the new combination of train-test sub-dataset for researchers to refer to in the future. Furthermore, the cost of each approach is different, and in many situations, accuracy is not the only metric the users consider. Other factors such as computing time, workflow robustness, effectiveness, stability, and reusability are all important in the approach selection processes in the real world. Each approach (the full-stack workflow rather than the chosen AI model) should be comprehensively considered before tossing it out and the difference of 1~5% in overall accuracy sometimes is acceptable by the eventual users. The full-stack workflow described in Section 3 has greatly helped us achieve the clean and reliable results, and we expect it to help clarify the mechanism and assist users to adopt this approach in vegetation mapping production.

Furthermore, 3D-1D-CNN has been already tested in another study with higher performance in handling hard-to-distinguish objects [26]. So, this paper serves as a validation experiment to prove that 3D-1D-CNN applies to the more general urban area classification, rather than only tree classifications. In conclusion, the 3D-1D-CNN model can still well mine spatial spectral information in a short time under the condition of small samples.

## 6. Conclusions

In this study, the 3D-1D-CNN was proposed to evaluate its performance in spectral-spatial information mining from hyperspectral data in a complex urban area, especially in the cloud shadow areas, and the following conclusions can be obtained.

1. The overall accuracy of the proposed 3D-1D-CNN is 96.32%, which is 23.96%, 11.02%, 5.22%, and 0.42% much higher than that of SVM, RF, 1D-CNN, and 3D-CNN, respectively. Although the results of 3D-2D-CNN and 3D-1D-CNN are similar, the time required for 3D-1D-CNN is shorter. The 3D-1D-CNN model can still well mine spatial spectral information in a short time under the condition of small samples.
2. 3D-1D-CNN has a strong ability to mine spectral-spatial information of cloud shadow areas, especially highway, commercial, grass_healthy, and grass_synthetic.
3. The combinations of spectral composition parameters, texture characteristics, and vegetation index can improve the classification accuracy. The optimal input size for training samples is $9 \times 9$ which achieved the highest overall accuracy of 96.32%.

In conclusion, 3D-1D-CNN, which achieves higher accuracy in a short time using small samples and fewer training parameters, is much better than 3D-CNN, RF, and SVM in complex urban areas, especially commercial, grass_healthy, grass_synthetic, and highway in the cloud shadow areas. Therefore, the 3D-1D-CNN can also be used in urban green space extraction using hyperspectral data.

**Author Contributions:** Conceptualization, X.M., Q.M. and X.Y.; Methodology, X.M., Q.M. and Z.Y.; Software, Z.Y.; Validation, X.M. and Q.M.; Investigation, Q.M.; Data curation, P.D., J.W. and C.L.; Writing—original draft, X.M. and Q.M.; Writing—review & editing, X.M., Q.M., P.D., J.W. and C.L.; Supervision, X.Y.; Funding acquisition, Q.M. All authors have read and agreed to the published version of the manuscript.

**Funding:** This research is funded by Natural Science Foundation of Shandong Province (NO. ZR2020QD019) and National Youth Science Fund Project of National Natural Science Foundation of China (NO. 42101337).

**Data Availability Statement:** The Houston data that support the findings of this study are openly available in [the IEEE GRSS Data Fusion Technical Committee for organizing the 2013 Data Fusion Contest.] at [https://hyperspectral.ee.uh.edu/?page_id=459, (accessed on 16 March 2022)].

**Acknowledgments:** The first author would like to thank the Hyperspectral Image Analysis group and the NSF Funded Center for Airborne Laser Mapping (NCALM) at the University of Houston for providing the data sets used in this study, and the IEEE GRSS Data Fusion Technical Committee for organizing the 2013 Data Fusion Contest. Thanks to the third author for providing the hyperspectral dataset of Surrey.

**Conflicts of Interest:** The authors declare no conflict of interest. The funding sponsors had no role in the design of the study; in the collection, analyses, or interpretation of data; in the writing of the manuscript, and in the decision to publish the results.

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
