# Peer review of "Urban Feature Extraction within a Complex Urban Area with an Improved 3D-CNN Using Airborne Hyperspectral Data"

_remotesensing, doi:10.3390/rs15040992_

Round 1
Reviewer 1 Report (Previous Reviewer 1)
This work is a new submission of a previous work on the use of a 3D-1D-CNN architecture for feature extraction and classification with hyperspectral data in urban areas. In comparison with the first submission many aspects of the article have been improved and in general I find that the authors have addressed all the concerns/questions that I made on the first submission. There are however a fe smal points that I think need to be addressed before publication:
- Table 4 and Table 5 provide the training parameters in the 3D-CNN and 3D-1D-CNN models. The difference comes only from the introduction of the conv1d_1 that adds more parameters but later reduces the number on the dense layer. The net result in the Houston dataset is that the number of training parameters are 198435 and 172011 respectively for the 3D-CNN and hte 3D-1D-CNN models. However, on Table 7 (Surrey dataset) the same model architecture are used but the number of parameters are different: 591579 and 172707. There are two questions. First why the number of parameters change between the 2 datasets? Second and more important, why the number of parameters for Surrey dataset in 3D-CNN is now ~3 times larger than 3D-CNN-1D?
- The text in section 5.5 after figure 18 perhaps needs some rework. It is not clear why the authors say that only the central pixel and some random pixels in the ROI in contrast to other studies wher eall pixels in ROI are used. The authors insist on the idea that the samples in other studies contain more pixels making the classification easier (actually the sentence is repeated, I would suggest to delete one of them). If that is the ase why not doing the same? Actually adding more pixels is not what was checked when using a larger input size in section 5.3? In Section 5.3 the authors discuss that actually it is better to use 9x9 what seems a contradict the other statement.
Author Response
Dear reviewer 1:
Thanks for your comments. Please find our response to the comments in the upload attachment.
Kind regrads!
Qixia Man

Reviewer 2 Report (New Reviewer)
1. On line 42 of page 1, The authors say, “The appearance of hyperspectral images at different times with high resolution makes it possible to overcome those problems with more details and improve the accuracy.” The examples given later do not mention which details were used , and how to deal with these details is not introduced in the article.
2. The expressions 54-56 of page 2 of the paper is not quite correct. The method mentioned above does not reflect the view that traditional classification methods cannot make full use of the rich spatial spectral information of hyperspectral data.
3. On page 9, line 234, SVM is used for binary classification. The principle of SVM used for multiclassification is not introduced in detail.
4. Surrey dataset cannot show the validity of 3D-1D-CNN. Why not use other publicly available data sets for verification.
5. Experiments comparing with other CNN algorithms should be added, and making tables to compare accuracy will be more intuitive.
6. To better illustrate the advantages of 3D-1D-CNN in improving the classification accuracy of shadow regions, it is suggested to increase the classification accuracy of 3D-1D-CNN with and without atmospheric correction and compare it with RF, SVM and some other algorithms of CNN.
7. Some references could be added in the introduction part, such as Doi: 10.1109/LGRS.2019.2945546, Doi: 10.1109/TGRS.2016.2593463, Doi: 10.1109/TGRS.2016.2584107
Author Response
Dear reviewer 2:
Thanks for your comments. Please find our response to the comments in the upload attachment.
Kind regrads!
Qixia Man

Round 2
Reviewer 2 Report (New Reviewer)
The author has addressed all the comments.
This manuscript is a resubmission of an earlier submission. The following is a list of the peer review reports and author responses from that submission.
Round 1
Reviewer 1 Report
This work studies the application of 3D-1D-CNN networks in complex urban areas. The classification results of this architecture are compared to those of 3D-CNN, SVM and RF. Although the overall accuracy achieved is in line with other results obtained with CNNs, it is also not the highest accuracy that can be achieved in this dataset with similar methods. In general, I am afraid this work does not show what is the novelty or the advantage of using this architecture over other possibilities. I find many other aspects that in my opinion show that this work is not ready for publication. These are the main ones:
- Results are mostly obtaiend from the Houston dataset. The comparison here with SVM and Random Forest is OK, but I miss comparison with any other CNN models that have been employed directly on this dataset. There are many, many of them on this very same journal.
- The OA achieved is reasonable, but it is not the best. Moreover, as the authors mention there are many classes not well classified. The detailed accuracy results per class shall be added to the results in Table 4 for comparison with other methods. There are other CNN methods that address the problem of imbalance on the method and do not seem to suffer from the problem with the shadows.
- Page 9 and 10 discuss the accuracy of the results and the authors provide explanations on why NDVI helps or the effect of image clipping on different classes. However, there is no proof on this and it would be as simple as repeating the analysis with different inputs or checking the accuracy of the classes with different clipping sizes. Instead of a guess, we could simply get the results to learn more about the method.
- The classification map in figure 7 wwith 3D-CNN or 3D-1D-CNN does not look as good as obtained with other methods in other publications where the city structures (building, roads) are better identifiable. Just to cite a couple of recent references just check:
1. Remote Sens. 2022, 14, 2215. https://doi.org/10.3390/rs14092215
2. Remote Sens. 2022, 14, 2265. https://doi.org/10.3390/rs14092265
- The Surrey dataset is introduced in the discussion section, the OA is lower than in the Houston dataset and it is mentioned the importance of spectral resolution. The difference in spectral resolution is between 9.6 nm in Surrey dataset and 4.9 nm in the Houston dataset. However the input of the CNN are not the spectra but the products (NDVI, MNF, PCA, GLCM) derived from the atmospherically corrected data. It is not clear how important would be that spectral difference in this case.
- Section 3.2 shall say which dataset is used (I guess Houston). Accuracy(5) is not well described, it is not clear if this is OA or AA from "5 results". Are the 5 results what is obtained from 5 different trainings?
- The authors use atmospheric corrections and show reflectance spectra, but there is no discusion on the effect of that process on the results. For instance, the atmospheric processing could be not very accurate and that affect the conclussions. Actually, some of the spectral show incorrect handling of atmospheric water vapour absorption, for instance in figure 6-right.
-In the shadows study, there are at least 2 (maybe 3) small areas which still contain shadows. That could affect the results. In general, I suggest to check the other studies with CNN on the Houston dataset that overcome naturally the problem with the shadows. Actually, it would be interesting to know if the problem is not coming from the architecture of the CNN but from the atmospheric correction and its handling of the shadows.
-Table 6 includes th eprecission recall F1 without cloud shadow, but for comparison the same shall be included for the complete dataset and the shadows-only dataset
- The loss function of the 3D-1D-CNN in the validation set shows much more variability with the epoch number than the 3D-CNN. The results seem less stable with epoch, while having less parameters to train. Why is that?
- Too many details explaining aspects that are pretty much standard these days like what is a 3D convolution or the size of the layers in each level of the network. In comparison, there are no details on how MNF, PCA, or NDVI are computed just a reference. If it is expected that the readers are aware of those details, in my opinion the same applies to how a CNN works except for the novelty of the 3D-1D architecture employed.
-Figure 9 does not explain what are the difference graphs
- The article is understandable, but the english grammar and some spelling error shall be revised.
Reviewer 2 Report
The authors in this paper deal with airborne hyperspectral data analysis. A new method of hyperspectral classification has been developed by incorporating three-dimensional convolutional neural networks (3D-CNNs).
The problem is not clearly stated in the abstract. The authors say, “However, how to mine and use this information effectively is still a great challenge.” What is the challenge here? Why to mine? Explain this, please.
The authors say “Firstly, several parameters were extracted from hyperspectral data.” What are those parameters? please mention them for the readers.
It is not clear why this study is conducted. How are the research problems handled?
The significant contribution of this work is missing.
Give reference to the dataset from where it is taken. Give a pictorial representation of the dataset in graph form. Give more samples of the dataset as figures.
Data pre-processing section is very week. The authors say “the minimum noise fraction (MNF) [22] and principal component analysis (PCA) [23] transforms are also employed to reduce or eliminate the Hughes phenomenon of hyperspectral data.” Please show the figure before and after applying these techniques
Figure 2 is not impressive it needs redrawing.
Section 3.2.2 is the main contribution in the paper and unfortunately it is very week. SVM and Random Forest are not explained in the paper even as small introduction.
Table 4 should also include the dataset column.
Strong technical contribution is missing. The problem and research gap is not described clearly, why this work is conducted. The present work is comparing with SVM, Random Forest and 3D-CNN but not actually with some past published work which shows this work is not competitive in the research.
The methodology applied in this paper is not discussed to its level. The core contribution is not effective.
Again, in conclusion section, the authors say “In conclusion, 3D-1D-CNN, which can cost a short time and achieve high accuracy with a few parameters,” these parameters are not discussed.